# Cachexia, a Systemic Disease beyond Muscle Atrophy

**DOI:** 10.3390/ijms21228592

**Published:** 2020-11-14

**Authors:** Elisabeth Wyart, Laure B. Bindels, Erica Mina, Alessio Menga, Serena Stanga, Paolo E. Porporato

**Affiliations:** 1Department of Molecular Biotechnology and Health Sciences, Molecular Biotechnology Center, University of Torino, 10126 Turin, Italy; elisabeth.wyart@unito.it (E.W.); erica.mina@unito.it (E.M.); alessio.menga@unito.it (A.M.); 2Metabolism and Nutrition Research Group, Louvain Drug Research Institute, UCLouvain, Université Catholique de Louvain, 1200 Brussels, Belgium; laure.bindels@uclouvain.be; 3Neuroscience Institute Cavalieri Ottolenghi, 10043 Orbassano (TO), Department of Neuroscience Rita Levi Montalcini, University of Turin, 10126 Turin, Italy; serena.stanga@unito.it

**Keywords:** cachexia, liver dysfunction, metabolism, inflammation, microbiota, anorexia, bone

## Abstract

Cachexia is a complication of dismal prognosis, which often represents the last step of several chronic diseases. For this reason, the comprehension of the molecular drivers of such a condition is crucial for the development of management approaches. Importantly, cachexia is a syndrome affecting various organs, which often results in systemic complications. To date, the majority of the research on cachexia has been focused on skeletal muscle, muscle atrophy being a pivotal cause of weight loss and the major feature associated with the steep reduction in quality of life. Nevertheless, defining the impact of cachexia on other organs is essential to properly comprehend the complexity of such a condition and potentially develop novel therapeutic approaches.

## 1. Introduction

Cachexia is a devastating syndrome related to the unleashed weight loss of the human body. It naturally occurs in the majority of severe diseases, including cancer, sepsis and major organ failure, including the liver, lung and kidney [1].

However, despite its huge prevalence, only relatively recently has cachexia become of main interest as we are beginning to define it as a set of specific molecular alterations, and therefore as a targetable condition.

For a long time, it has been considered only as an epiphenomenal complication occurring in the terminal phase of disease. However, the presence of alterations promoting cachexia at earlier stages is progressively emerging [2].

Still, only in the last ten years has a consensus formed on the clinical definition of cachexia, i.e., the unwanted loss of at least 5% of lean mass in the previous six months [3]. Hence, by definition, cachexia is a syndrome affecting skeletal muscle mass and function. Skeletal muscle wasting being the most evident component of cachexia, it is not surprising that it represents the main focus of interest incachexia research. Moreover, striated muscle wasting is also a direct cause of death because it causes cardiac dysfunction, and indirectly causes respiratory distress by impacting respiratory muscle function [4]. Indeed, many cancer patients experience severe cardiac abnormalities such as cardiac atrophy, remodeling, and dysfunction. Additionally, those symptoms are frequently worsened by chemotherapeutic treatment warranting the rapid development of the cardio-oncology research field.

Altered mitochondrial metabolism is an important component of the wasting process, as dysfunction in mitochondrial function precedes the development of atrophy [5,6], an issue present also in neurodegenerative diseases. Indeed, in neurodegeneration the loss of specific neuronal populations and circuits is triggered by mitochondria dysfunctions [7,8]; furthermore, the progressive motor decline is related to the fact that mitochondria-enriched muscles are also impacted by their dysfunctions.

An important element which has been uncovered is the impact of altered fatty acid metabolism in the progression of muscle wasting, linking increased fatty acid oxidation, into the muscle, to the initial step of cachexia [9,10,11]. Consequently, an important line of research has been also initiated with the definition of the molecular and metabolic alterations occurring in fat tissue, most notably concerning browning and increased energy expenditure [12].

Nevertheless, cachexia is much more than a syndrome characterized by unwanted weight loss, as cachexia is a syndrome affecting the function of multiple organs and therefore the entire body as a physiological system (Figure 1). This characteristic of the syndrome is why cachexia is named as such—the word derives from Greek, meaning ‘bad condition’.

Importantly, understanding how the various tissues are impacted by the cachectic condition and how the dysfunction of various organs might contribute to the progression of the disease is an essential element to clearly identify drivers of this terrible condition and ultimately to provide novel therapeutic and diagnostic options.

## 2. Immune System

Systemic inflammation and cytokine storms are key drivers of cancer cachexia [13]. Numerous pro-inflammatory pathways are activated by the tumor mass and are generated through a strict crosstalk between stromal cells and immune system. These signaling molecules act both centrally, by controlling the central nervous system, appetite, energy intake and expenditure, and peripherally, by promoting catabolism in target organs such as skeletal muscle and adipose tissue [14,15]. Findings have shown that metabolic and molecular changes in skeletal muscle, linked to immune dysfunction and systemic inflammation, already occur in patients before any evidence of body weight loss [14]. Consequently, it is clinically relevant to investigate the onset of immune suppression, identify early risk factors and counteract immunity dyshomeostasis in the course of the cachexia process, in order to improve the quality of life [16]. The dysfunction of the immune system ultimately leads to susceptibility to infections, and therefore, a poor clinical outcome [14]. The first cytokine held responsible for causing anorexia-cachexia syndrome was tumor necrosis factor alpha (TNF-α), initially identified as “cachectin” [17]. TNF-α is released by activated macrophages, CD4+, neutrophils and eosinophils in patients with various types of cancer [16]

Human studies highlighted that TNF-α, through nuclear factor kappa-light-chain-enhancer of activated B cells (NFkB) activation, triggers the ubiquitin–proteasome pathway and induces nitric oxide species (NOS) production and skeletal muscle degradation [18,19,20]. Furthermore, TNF-α induces adipose tissue wasting through the inhibition of lipoprotein lipase (LPL), promotion of lipolysis and upregulation of UCP1 and UCP2 [21,22,23]. However, the action of TNF-α can only be explained in the context of concomitant presence of other cytokines [24]. Produced mainly by macrophages, Interleukin 1 alpha (IL-1α) is able peripherally to inhibit LPL activity and stimulate lipolysis in adipocytes [25]. Centrally, IL-1α is able to suppress appetite and to induce anorexia by increasing plasma concentrations of satiety-drivers tryptophan and serotonin [26], and by blocking neuropeptide Y (NPY) secretion [27]. Released by macrophages, Interleukin 1 beta (IL-1β) has been, together with TNF-α, one of the most clinically studied cytokines, as well as being also better associated with anorexia, weight loss and sarcopenia than Interleukin 6 (IL-6) [28,29,30,31]. IL-1β decreases food intake and body weight via leptin activation in adipose tissue, but it is also able to induce anorexia independently from leptin, by induction of the melanocortin system in the hypothalamus (the main area in the brain regulating feeding behavior and body weight). Furthermore, peripheral IL-1β can exacerbate the loss of appetite by inhibiting gastric emptying and motility [32]. Experimental evidence in animal models suggests that interferon gamma (IFN-y), produced by activated T and NK cells, is implicated in the loss of body weight and atrophy of adipose tissue [33,34,35]. Produced by macrophages but secreted mainly by tumor cells, IL-6 is able to increase muscle cells autophagy in murine models of cancer [36,37] and affect cachectic patients’ body composition by targeting adipose tissue, gut, and liver tissue [37,38].

Studies have suggested that the development of cancer cachexia might be linked to the expansion of immature myeloid populations. Myeloid derived suppressor cells (MDSC), immature myeloid cells in various stages of differentiation, play a key role in the overproduction of cytokines and inflammatory mediators, which might contribute to altered fat metabolism and body wasting [39]. Functional studies in cancer patients have found a positive relationship between immune cell infiltration (granulocyte/phagocytes, and CD3−CD4+ cells) and muscle mass status, whereas a negative correlation has been established between CD8 T cells and muscle catabolic pathways [40]. In a murine HCC (Hepatocellular Carcinoma) model, a decreased macrophage infiltration in visceral tissue has been associated with the loss of adipose tissue [39,41], whereas in clinical studies, activated macrophages have been found to be infiltrated in fat. To date, the mechanisms by which macrophages modulate adipocyte function are still unclear [42,43,44,45]. It is clear that tissues and organs directly involved in the cachectic process, such as adipose tissue, and the brain, liver, gut, bones and heart, are connected to immune system. TNF-α, IL-1α/β and IL-6 released by stimulated macrophages are responsible for systolic heart failure [46] and liver dysfunction with consequent insulin resistance [34], cholestasis [35] and steatosis [36]. The same cytokines together with the receptor activator of nuclear factor kappa-Β ligand (RANKL) can induce bone loss by apoptosis of osteoblasts and the differentiation of osteoclast precursor cells (OCPs) into activated bone resorbing cells [47]. Immune cell infiltration and other inflammatory changes can perturbate the intestinal microbiota composition and exacerbate mucosal damage and gut permeability [48]. Studies on murine models of cancer cachexia have demonstrated that neutrophils infiltration and microglia activation can contribute to brain dysfunction, probably through affecting sympathoexcitatory and anorexigenic neurons [49,50]. Multiple mechanisms are involved, therefore the impact of the immune system and its role in cancer cachexia are extraordinarily complex, underexplored and need to be deepened [14]. Bearing in mind the involvement of the immune cells and their cytokines in cachexia, the development of therapeutic strategies has focused on counteracting their action pharmacologically [51]. In a clinical trial on advanced pancreatic cancer patients, the drug thalidomide (a-N-phthalimidoglutaramide) was used as a TNF-α production suppressor with promising results on body mass preservation [52]. Other anti TNF-α strategies (etanercept and infliximab) were used in clinical trials to improve the symptoms of cachexia, the first one successfully whereas the second one not [53,54]. Some studies have demonstrated the efficacy of blocking IL-6 pathway by monoclonal antibodies in murine models of cancer cachexia (tocilizumab) and in non-small-cell lung carcinoma (NSCLC)-patients (ALD518), leading to an improvement of muscle mass and fitness [55,56]. Furthermore, broad-spectrum peptide immunomodulator drugs which modulate cytokine action have been evaluated in human clinical trials on cachectic patients (not only cancer related), resulting in a good safety profile and improvement of body weight and physical performance [57]. Because cachexia is a multifactorial and multiple organ dysfunction syndrome, a better understanding of the role of the immune system in inter-tissue crosstalk and a multimodal approach are essential for the design of effective therapeutic strategies.

## 3. Liver

As described above, proinflammatory cytokines are increased in cancer cachexia and liver function is strongly affected by them. Animal models of cachexia revealed a large number of liver alterations: proinflammatory cytokines induced an increase in inflammatory mediators in the liver, mainly by Kupffer cells, leading to insulin resistance [58], cholestasis [59] and steatosis [60]. Tumor-induced inflammation has been shown to promote the alteration of liver circadian homeostasis, altering AKT, AMPK and SREBP signaling, as well as insulin and glucose homeostasis [61]. The secretion of cytokines in the liver is known to influence the expression and the function of bile transporters determining the accumulation of the bile in the liver [59,62], which results in liver damage [63].

On the other hand, the liver contributes to cancer cachexia because it is involved in the acute phase response (APR) to tissue injury and inflammation by synthesizing acute phase proteins, such as fibrinogen or serum amyloid A (SAA). In cachexia, the production of APR proteins is strongly upregulated both in the C26 model [64] and in cancer patients [65]. Interestingly, it has been directly linked to an increase in resting energy expenditure in pancreatic cancer patients [66]. It is likely that to allow a rapid and efficient synthesis of APR protein, skeletal muscle undergoes a catabolic process to mobilize and supply the liver with the necessary amino acids [64]. Therefore, this mismatch in amino acid content between skeletal muscles and APR proteins can contribute to muscle atrophy [67]. This process is mediated by the activation of the IL-6-STAT3 signaling pathway. Moreover, due to the liver being a controller of whole-body energy expenditure, it also plays an important role in the etiology of cachexia. It has long been known that tumor ATP production heavily relies on glycolysis, thus cancer cells consume high amounts of glucose and release high levels of lactate [68]. An increase in Lactate dehydrogenase A chain (LDHa) and in the lactate transporter MCT1 was found in the liver of colon 26 carcinoma model (C26) mice, suggesting a strong alteration of the hepatic lactate metabolism [69]. Indeed, high circulating lactate gives rise to the “Cori cycle” in the liver: hepatic cells reconvert circulating lactate into glucose through gluconeogenesis. This interorgan lactate cycling is a very inefficient metabolic process resulting in a negative energy balance. Nevertheless, studies have not found significantly altered lactate levels in the liver [70], thus requiring further experiments of metabolic tracing to measure the possible degree of the Cori cycle. Lactate is not the only substrate fueling hepatic gluconeogenesis, and it was postulated that amino acids mobilized from the catabolism of skeletal muscle can serve as an alternative source for energy production and sustain liver gluconeogenesis, further contributing to energetic inefficiency [71,72]. Moreover, similarly to adipose tissue, it was suggested that uncoupling of mitochondrial oxidative phosphorylation also occurred in cachectic livers. Indeed, cachectic livers from C26 tumor-bearing mice were found to present a reduced respiratory control ratio (an index of oxidative phosphorylation (OXPHOS) coupling efficiency) and an elevated proton leak respiration generating an increase in hepatic heat production, and therefore further contributing to the increased resting energy expenditure and weight loss [73]. In parallel, a cachectic liver results in the reduced export of fatty acids [74], further contributing to dyslipidemia in cachexia.

Ketogenesis is another hepatic metabolic process altered in lung and pancreatic cancer cachexia models [72,75]. Ketone bodies are produced from fatty acid oxidation in the liver and can be used for energy production in the skeletal muscle, heart or brain, especially in the case of starvation. Unexpectedly, serum levels of ketones were low in a lung mouse model of cancer cachexia, despite the strong decrease in food intake [72]. Reduced ketogenesis (likely induced by IL-6) impedes the physiological response to low food intake and prevents an efficient systemic energy production [75]. Consequently, low ketogenesis in cachexia further exacerbates the energetic crisis in cachexia, and results in elevated glucocorticoid levels triggering both a strong catabolic program and an anti-anabolic program in skeletal muscle, ultimately leading to atrophy [75,76]. Targeting ketogenesis using a PPARa agonist (fenofibrate) resulted in less circulating glucocorticoid and therefore a reduced muscle atrophy in a cancer cachexia mouse model [72], highlighting this complex cross-talk between liver, skeletal muscle and glucocorticoids in which glucocorticoids modulate hepatic function and induce skeletal muscle protein degradation in order to enhance gluconeogenesis.

Collectively, a growing amount of evidence indicates that liver dysfunction and skeletal muscle degradation are intrinsically linked and contribute to tumor progression in cachexia. However, more research is needed to determine whether targeting hepatic metabolism could improve cachexia status in cancer patients.

## 4. Brain Dysfunction and Neuroinflammation

The brain, mainly via the hypothalamus, is the master regulator of systemic energy homeostasis [77], making it an important player in the etiology of cachexia. Peripheral metabolic signals from the liver, adipose tissue, pancreas and skeletal muscle are integrated in the hypothalamus, where populations of specialized neurons coordinate the response to altered metabolic conditions by driving changes in energy expenditure and food intake.

Anorexia is a frequent co-morbidity of cancer cachexia and considerably contributes to the negative energy balance observed in cachectic patients [78]. Appetite regulation is a complex process involving a broad variety of signals (hormones, nutrients, neuronal) converging to the hypothalamus. Those signals, either orexigenic (stimulating food intake, such as ghrelin) or anorexigenic (inhibiting food intake, such as leptin, insulin cholecystokinin, peptide YY, glucagon-like peptide 1, pancreatic polypeptide) stimulate distinct neuronal populations. Orexigenic signals stimulate neurons expressing NPY and agouti-related peptide (AgRP), while anorexigenic hormones stimulate neurons expressing the cocaine and amphetamine-regulated transcript (CART) and pro-opiomelanocortin (POMC) [79].

There is consistent evidence that increased hypothalamic inflammation is involved in the disruption of homeostatic regulation of appetite [80]. The privileged connection of hypothalamus to peripheral circulation (via the hypophyseal portal system) makes it very sensitive and reactive to the massive secretion of inflammatory cytokines occurring in cachexia. In addition to directly promoting muscle wasting and lipolysis, cytokines such as IL-1b and TNF-α have been shown to strongly decrease food intake when injected intracerebroventricularly in rodents [81,82]. Interestingly, blockades of TNF-α signaling with neutralizing antibodies, or administration of an IL-1b receptor antagonist were both able to prevent anorexia [83,84]. Mechanistically, pro-inflammatory cytokines such as IL-1b have been shown to overstimulate anorexigenic POMC neurons [85]. Similarly, the tumor-derived Leukaemia inhibiting factor (LIF), identified as a driver of cancer cachexia [86] activates anorexigenic POMC neurons [87]. Consistently, targeting the melacocortin system through central administration of an antagonist to the melanocortin 4 receptor (MCR4) appeared to be an efficient strategy for counteracting anorexia and cachexia in a murine cancer model [88,89].

Although the anorexigenic circuit is overactivated in cachexia, contrarily, the orexigenic axis appears to be dysfunctional as suggested by the decreased level of circulating NPY in anorexic cancer patients [90]. Coherently, alterations in the NPY system were also found in a rat model bearing a methylcolantrene)-induced sarcoma as the expression of the NPY receptor was decreased in the hypothalamus [91]. Interestingly, tumor resection was sufficient to restore the hypothalamic expression of NPY [92,93]. It has been suggested that increased levels of serotonin were responsible for the inhibition of hypothalamic NPY secretion [93,94]. It is likely that elevation of ghrelin levels is a compensatory response to cancer-induced anorexia [94,95], however it is not sufficient to restore appetite and reveals a mechanism described as “ghrelin-resistance” [96]. Finally, another emerging factor contributing to anorexia is the growth differentiation factor 15 (GDF15), which has been linked to cancer-induced emesis [97].

Decreased food intake is an important feature of cachexia, however nutritional strategies aiming to reverse anorexia turned out to be inefficient for preventing body weight loss [98]. Although these disappointing results might be explained in part by a lack of rigorous randomized controlled clinical trials, they also suggest that decreased energy intake is not the sole cause of cachexia. Indeed, nutritional support does not address the underlying catabolic state, and therefore may be of limited efficacy. In addition to appetite regulation, the hypothalamus is also involved in energy expenditure regulation, in particular through the control of thermogenesis. About 50% of cancer patients are hypermetabolic, meaning that they have an elevated Resting Energy Expenditure (REE) [99], an issue that has been associated with shorter survival in metastatic cancer patients [100]. Neuroinflammation appears to be a driving element of hypermetabolism. In particular, it has been shown that increased TNF-α in the hypothalamus triggers heat production through β3 adrenergic signaling to brown adipose tissue, therefore contributing to energy loss [101].

In addition to its effects on food intake and thermogenesis, the hypothalamus can also contribute to skeletal muscle catabolism through the hypothalamic–pituitary–adrenal (HPA) axis. Several studies have reported elevated circulating glucocorticoids in both animal models of cancer cachexia and cancer patients [72,75,102]. Glucocorticoids are well known inducers of skeletal muscle atrophy [103] and their secretion by the adrenal gland is placed under the control of the HPA axis, which can be stimulated by pro-inflammatory cytokines such as IL-1β, leading to the secretion of glucocorticoids by the adrenal gland ultimately promoting skeletal muscle and adipose tissue wasting [81].

The study of food intake regulation in cachexia led to the development of different appetite targeting molecules producing promising results, such as Anamorelin [104]. Nevertheless, the knowledge about the implication of other hypothalamic functions, such as the control of energy expenditure, is still scarce, and future investigations are greatly needed to fully unravel the role played by the brain in the development of cachexia.

## 5. Alterations of the Intestinal Barrier and the Gut Microbiota

Beside the tumor, another source of inflammation may come from the translocation of pathogen-associated molecular patterns (PAMPS) such as lipopolysaccharides (LPS) and peptidoglycans, arising from the gut microbiota. Gut barrier dysfunction has been described in several mouse models of cancer cachexia [105,106,107], with an alteration of the tight junctions, an increased gut permeability, morphological changes, a local immunosuppression, and a decreased expression in antimicrobial peptides. In the C26 model, this alteration in the gut barrier function was attributed to the increased systemic levels in IL-6 [108] and fostered by the emergence of specific opportunistic bacteria such as *Klebsiella oxytoca* [109]. Interestingly, LPS has been shown to alter muscle progenitor cells in mouse and chicken embryos [110] while directly promoting muscle catabolism in adult mice through the Toll-like Receptor 4 [111].

Alterations of the gut microbiota composition itself has been described in several mouse models of cancer with cachexia [105,112,113,114]. Targeting these alterations using selected prebiotics and/or probiotics led to a reduced accumulation of tumor cells in the liver, reduced muscle atrophy, improved morbidity and/or sparing of the adipose tissue [112,113,115]. In these experiments, a modulation of the systemic inflammation was pinpointed as the likely mechanism for the reduced muscle atrophy observed upon dietary modulation of the gut microbiota composition. However, other mechanisms may explain the impact of the gut microbiota on muscle metabolism. As a first example, phenolic compounds produced through microbial transformation have been shown to induce muscle glucose transport [116] and to foster muscle hypertrophy in vivo [117]. In addition, bacterial metabolites such as short-chain fatty acids and bacterial-host cometabolites such as bile acids may also affect muscle function and metabolism [118]. For instance, administration of short-chain fatty acids to microbe-free mice improved grip strength [119], and the fibroblast growth factor 19 (FGF19), an intestinally produced hormone controlled at the transcription level by bile acids, increased muscle mass in healthy mice [120]. Thibaut et al. have shown an alteration of bile acid pathways in mouse models of cachexia as well as in a cohort of colon cancer patients [121]. Finally, quorum sensing molecules, traditionally only seen as intra-bacterial communication molecules, have been shown to affect parameters such as viability, differentiation, inflammation, mitochondrial changes and protein degradation in C2C12 myotubes [122].

Noticeably, most of these findings have been made using in vitro approaches and/or mouse models. Whether these observations are clinically relevant remains to be determined. Interestingly, Costa et al. reported morphological and inflammatory changes in the intestinal mucosa in healthy tissues of cachectic colon cancer patients compared to weight-stable colon cancer patients [48]. A full characterization of the gut microbiota of leukemia patients with and without cachexia is currently ongoing in an academic multi-centric prospective study (NCT03881826) and it should bring more light on this topic.

## 6. Insulin Resistance

Disturbed glucose metabolism was recognized as early as 1919 in cancer patients [123]. Insulin resistance has been observed in patients with various tumor types and correlates with a higher risk of mortality [124]. Importantly, the causal role of the tumor in promoting insulin resistance was confirmed in studies where glucose tolerance was restored after surgical removal of the tumor [125]. A study comparing glucose tolerance in cachectic cancer patients and non-cachectic cancer patients revealed that both groups had a lower glucose uptake rate than healthy controls, but the severity of glucose intolerance was greater in cachectic cancer patients suggesting its potential implication in the development of cachexia [126]. Decreased insulin sensitivity was also confirmed in several animal models of cachexia, such as C26 tumor-bearing mice [127] and Walker 256 tumor-bearing rats [128] but also in drosophila [129]. Asp et al. [127] investigated whether cachexia was a cause or a consequence of insulin resistance in cancer; they concluded that in C26 tumor-bearing mice, insulin resistance is an early event preceding the onset of skeletal muscle atrophy and that treatment with the insulin sensitizer rosiglitazone alleviated early cachectic features [127]. Accumulating evidence identifies inflammation as a potential cause of insulin resistance in cancer. Pro-inflammatory cytokines such as TNF-α can inhibit insulin signaling and insulin receptor activation [130]. They can also directly trigger dysfunction and apoptosis of pancreatic β cells leading to impaired insulin secretion [131].

The implications of the insulin resistance for the cachectic phenotype are numerous. Firstly, insulin plays a major role for skeletal muscle mass regulation, because it is a potent anabolic factor inhibiting muscle proteolysis and enhancing protein synthesis [132]. Therefore, insulin resistance might directly promote muscle wasting. Coherently, insulin resistance is present in many other catabolic diseases involving muscle loss such as AIDS [133], diabetes mellitus [134] and chronic heart failure [135].

Insulin resistance is known to activate hepatic gluconeogenesis in various chronic diseases, including cancer. This abnormal increase in de novo glucose production contributes significantly to the elevated resting energy expenditure associated with cachexia. Indeed, hepatic glucose production correlates with weight loss severity in patients with colorectal carcinoma and lung cancer [136,137]. This increase in gluconeogenesis might also be explained by the increase in glucagon level observed both in humans and animal models of cachexia [138,139].

Finally, the tumor itself might benefit from systemic insulin resistance and glucose intolerance. Increased insulin secretion, which occurs in the early phase of insulin resistance, promotes tumor growth per se [140]. Moreover, the combination of impaired glucose uptake from skeletal muscle and adipose tissue with the increased gluconeogenesis in the liver favors the tumor’s access to glucose and consequently tumor growth, as elegantly demonstrated by Ye et al. in 2018 [141]. The release of amino acids (due to excessive muscle catabolism) and lactate (due to the high proliferative rate of the tumor) foster this vicious cycle by fueling the Cori cycle, further promoting tumor growth and subsequent wasting [123,142].

## 7. Bone Density

Bone mineral is the second largest lean tissue compartment after skeletal muscle, and it stores minerals, collagenous proteins, growth factors and cytokines [143]. Cancer represents a major risk factor for bone loss and fracture. It has been reported that lung cancer patients with 30% of body weight loss showed lower mineral content than the control group [144], and that advanced cancer patients with skeletal muscle loss often have bone metastases, bone pain and hypercalcemia [145]. Beside these observations, bone loss has been poorly studied in cachexia, even though skeleton decay can further diminish cancer patients’ quality of life. Interestingly, some pathways of skeletal muscle mass regulation are common with bone loss and osteoporotic signaling. In particular, it has been shown how different conditions such as aging, reduced biomechanical loading and systemic inflammation (like in cancer), can lead to concomitant bone and muscle loss through activation of the NF-kB signaling pathway in various murine models [146]. In addition to wasting pathways acting in parallel, there is a growing body of evidence showing how bone- and muscle-derived cytokines (osteokines and myokines, respectively) can reciprocally influence each other in a dynamic crosstalk. For instance, osteocalcin is produced in response to insulin by primary murine osteoblasts, promoting a positive feedback by increasing insulin synthesis and, more importantly, its sensitivity in adipose tissue and skeletal muscle in vivo [147,148]. In addition, osteocalcin signaling has been shown to increase mitochondrial content in skeletal muscle of osteoblast-specific *Esp*-deficient mice [149]. Coherently, it was seen that levels of active osteocalcin positively correlate with increased lower-limb strength in a human study of older women [150], further confirming the evidence of a metabolic crosstalk between bone and the skeletal muscle.

In contrast, other osteokines induce atrophy or decrease skeletal muscle function, such as fibroblast growth factor 23 (FGF-23), activin, and Transforming Growth Factor beta (TGFβ) [151,152,153]. On the other side, skeletal muscle secretion of factors, such as fibroblast growth factor 21 (FGF21), can promote bone resorption [154], as well as muscle wasting [155] in FGF21 transgenic mice. Beside the knowledge of these mediators, their action in cancer cachexia is poorly understood.

Several murine models of cancer cachexia present bone alterations. Experimental models of colon cancer cachexia exhibited different levels of bone loss. C26 was characterized by extensive body weight and muscle loss, but moderate bone depletion and no alterations in bone strength. HT-29 and Apc^Min/+^ showed significant reduction in whole-body bone mineral density (BMD) and content (BMC), and in trabecular bone volume fraction, number and thickness; only Apc^Min/+^ mice had a significant decrease in bone strength [156]. In a murine cachexia model of Lewis lung carcinoma (LLC), BMD was reduced by about 5% and 6% at 21 and 25 days post tumor cell inoculation, respectively [157]. Besides the descriptive nature of these studies, bone-muscle mediators were not extensively studied. Differently, a study on pancreatic cancer cachexia in mice, underlined the role of TGFβ in bone and skeletal muscle loss [158]. Two models of pancreatic ductal adenocarcinoma (PDAC), namely KPC and Pan02, underwent a significant decrease in body weight, bone mineral content, and density. Interestingly, TGFβ inhibition improved cachectic mice survival, body weight and bone mineral density. Nevertheless, the specific effect of a TGFβ blockade on skeletal muscle was not assessed [158].

A better characterized aspect of bone-muscle crosstalk is the effect of osteolytic bone metastases on skeletal muscle. Tumor cells can destroy the physiological remodeling of bone, producing factors that activate osteoclasts and induce bone resorption and the release of factors (i.e., TGFβ) that sustain tumor growth and osteolysis in a vicious cycle [159]. Particularly, the massive release of TGFβ in mouse models of human osteolytic prostate, lung and breast cancer metastases caused a systemic skeletal muscle weakness [160]. TGFβ released from bone metastases induced the overexpression of the NADPH oxidase 4 (Nox4), leading to oxidative stress and ryanodine receptor (RyR1) oxidation, ultimately causing calcium leaking and skeletal muscle weakness in mice bearing osteolytic breast cancer cell line 4T1 [161].

Despite cachexia often being associated with advanced cancer patients with bone metastases, mechanisms of bone-muscle crosstalk or bone loss in cachectic models without osteolytic metastases are still scarcely understood and deserve more attention.

## 8. Conclusions

Cachexia is an illness with a terrible impact, affecting both quality of life and the availability of therapeutic options. Hence, there is a growing awareness of this multifactorial syndrome, which will certainly promote earlier diagnosis and improved management. Importantly, we speculate an increase in the future number of cachexia cases, along with increased awareness and the improvement of diagnostic tools. Therefore, a complete comprehension of all the aspects linked to cachexia will be pivotal in developing proper therapeutic options to prevent the typical weight loss and the general state of asthenia and frailty. To date, the vast majority of experimental research on cachexia and the definition of the drivers of this process is based on the identification of the molecular mediators of some specific aspects of cachexia, such as the ones resulting in skeletal and cardiac muscle wasting and lipolysis. As shown in Figure 1, only some organs are emerging as impacted by cachexia, but this is just the tip of the iceberg, which includes mostly skeletal muscle, body fat stores and, with the heart and immune system only partially affected. Although the link to cachexia of several other organs is evident, as discussed in this review, the knowledge of the effects linking cachexia to those tissue is still rather limited. Surprisingly, we still have limited knowledge of the impact of cachexia on other tissues and on the impact of other organ dysfunction on the global repercussions on body weight loss and frailty. For instance, this is particularly evident in the liver, being an essential organ driving systemic body homeostasis. Surprisingly, hepatic function is almost neglected in the field, with few research papers focused on secondary hepatic dysfunction in cachexia. Nevertheless, an important target deserving further investigation is the circadian rhythm alteration [61] which has been proposed to foster altered insulin sensitivity, a hallmark of cachexia. The modeling of such alteration will be pivotal in providing better therapeutic options, as well as identifying the possible ties with general wasting. Surprisingly, despite the importance of such a disease, there are still several organs and cell types which are almost neglected, such as many immune cell populations. Importantly, another issue relates to the use of a limited number of murine models (mostly C26 injected cells), based on the concept that a single type of cachexia exists. However, given the different origin of cancer cachexia, simply because it might be related to different organs, this assumption is rather reductive. Hence, a better definition of the crosstalk between organs will be pivotal for the understanding of cachexia from a molecular standpoint and identifying the real drivers.

## Figures and Tables

**Figure 1 ijms-21-08592-f001:**
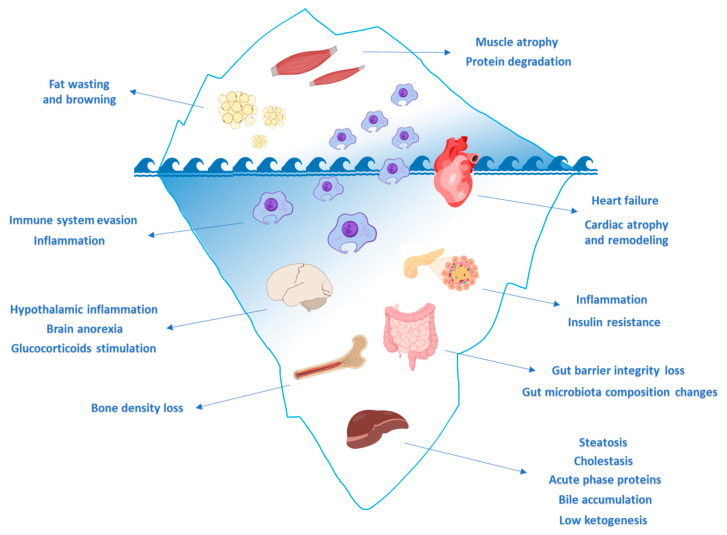
Muscle cachexia is the tip of the iceberg. A growing body of evidence clearly indicates that cancer-induced muscle atrophy is only the tip of the iceberg. Indeed, multi-organ dysfunctions are parallelly ongoing during tumor growth and, in turn, their dysfunction is promoting muscle wasting in cachexia. Created with BioRender (https://biorender.com) and SciDraw (https://scidraw.io) software.

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
