# Peer review of "Cachexia, a Systemic Disease beyond Muscle Atrophy"

_ijms, 2020, doi:10.3390/ijms21228592_

Round 1
Reviewer 1 Report
The authors reviewed the literature and summarized the pathological impact of cancer cachexia on six organs/systems other than skeletal muscle. Considering that cachexia is a multifactorial disease, their review approach is reasonable and unique. The choice of organs/systems is appropriate, and literature was adequately cited and updated in each section. However, the description mixed data from experimental models and that from human study. Discussions in each organ/system do not related to one another without specific clinical implications. The conclusion was unremarkable, with a few original messages. We need new insights or original thought after integrating all the pieces of evidence the authors gathered.
Accordingly, I think this manuscript needs several revisions before acceptance. Thanks.
Comments
1) Please separate clinical data and experimental data in each section.
2) I recommend the "clinical implication" part in each organ/system section.
3) I recommend adding a discussion section to integrate all the information and show the authors' original thoughts obtained from this review process. We also need a limitation section inside the discussion.
4) Authors did not use Figure 1 in the main text. Please explain Figure 1 in detail in the new discussion section.
Author Response
We thank the reviewer for the constructive comments and for evidencing some critical weaknesses. We believe we addressed those comments and we provide inhere an itemized list of modifications, as requested by the journal policy we include also the line number of changes performed (based on the file with “track-changes” included)
Comments
1) Please separate clinical data and experimental data in each section. 2) I recommend the "clinical implication" part in each organ/system section.
We included clearer indication when studies were available in clinical or experimental settings (e.g. line 75; 85; 103-107; 124-136;250; 264-271;306-307; 364). In this way, it is more evident the clinical implications related to each tissue dysfunction. Nevertheless, data are still very limited in clinical settings, an issue which has been consequently discussed more at the end of the paper (see further comment)
3) I recommend adding a discussion section to integrate all the information and show the authors' original thoughts obtained from this review process. We also need a limitation section inside the discussion.
Based on his comments we basically rewritten the discussion (Line 401-425) in a more incisive way, insisting on the under-investigated aspect of cachexia and proposing some priorities (i.e. liver and inflammation) important for further research. One major limitation we highlighted is the impossibility of drawing general conclusion on organ failure in cachexia as clinical data are still very limited, while animal model is mostly focused on a specific type of cancer.
4) Authors did not use Figure 1 in the main text. Please explain Figure 1 in detail in the new discussion section.
We cited and discussed the figure and included in the discussion (line 408-412)
Reviewer 2 Report
General comment: The manuscript provides an insightful orientation the concept of multiple tissues and physiological systems being involved in the complex syndrome of cachexia. Provision of more sentences throughout the manuscript that provide synthesis, critical evaluation and insight from the authors perspective would be good in order to distil the meaning of the how the examples given integrate and operate as a system to the audience. This would greatly strengthen the piece.
- Introduction
Line 22: suggested 'failure' rather than failures
Line 26: 'did cachexia become of main interest'
Line 33: recommended to read...'Skeletal muscle being...
Line 36: clarify further as they are two different types of muscle - is the idea about diaphragm or intercostal muscles - being striated
Line 38: is there a trigger for mitochondrial dysfunction to instigate atrophy? or is it a metabolic issue? please clarify further and make the link for the reader. what is the commonality with neurodegeneration? is the mechanism similar? Please give insight into the molecular and cellular similarities and differences.
Line 45: re-connect the molecular link here between BAT-SkMUS for the audience and its relevance for providing a model of tissue, cellular and molecular cross-talk in the cachectic condition.
Recommend adding more to the sentence: …. function of multiple organs 'and therefore the entire body as a physiological system'
Line 49: recommend a change of term from ‘and in fine to provide’ to ‘and ultimately to provide' to improve sentence flow
- Immune System
Line 56: recommend that (9, 10) citations are placed in one set of brackets
Line 60: 'of the cachexia'
Line 62: Remove the space between ‘(9). and The’
Line 64: Remove the space between ‘(12). and TNF-a’
Line 70: recommend that (14, 15, 16) citations are placed in one set of brackets
Line 77: use full name for IL-6 then acronym in brackets; recommend to use IL-6 (21, 22, 23) citations in one set of brackets
Line 81: recommend using IFN-y
Line 84: recommend that (25, 26, 27) citations are placed in one set of brackets
Lines 85-86: recommendation that the transignaling between tissues concept should be brought back here, with more discussion of molecular and cellular linkages
Line 92: recommend that (30, 31) citations are placed in one set of brackets
Line 100: is connection of immune cells to various tissues a feature? Can the authors expand on this concept? It is also represented this way in Figure 1.
- Liver
Line 107: remove full stop ‘homeostasis. (37).’
Line 139: recommend that (47, 50) citations are placed in one set of brackets
Line 142: recommend using IL-6
Line 145: recommend that (50, 51) citations are placed in one set of brackets
Line 146: does this suggest cross-talk with the GR? Please elaborate.
- Brain dysfunction and neuroinflammation
Lines 151-152: include citation to support this claim
Lines 169, 171, 173: recommend using IL-1b
Line 170: recommend that (55, 56) citations are placed in one set of brackets
Line 172: recommend that (57, 58) citations are placed in one set of brackets
Line 178: recommend that (62, 63) citations are placed in one set of brackets
Line 181: state which rat model explicitly
Line 182: two concepts presented here, it is recommended that they should be separated out for the reader to understand the connections between mouse models and patient studies more clearly. what is the outcome of all of these findings....does this suggest further complexity and possible networking? lead the reader into your interpretation of this relevance of these molecules.
Line 184: font change here for the word ‘Cachexia’
Line 192: more critical evaluation as to why this happens is recommended here?
Line 197: remove space after TNF-a
- Alterations of the intestinal barrier and the gut microbiota
Line 210: recommend that (76, 77, 78) citations are placed in one set of brackets
Line 212: C26 should be introduced in the beginning as an acronym for colon 26 carcinoma model
Line 210: recommend that (76, 77, 78) citations are placed in one set of brackets
Line 218: recommend that (76, 83, 84, 85) citations are placed in one set of brackets
Line 220: recommend that (83, 84, 86) citations are placed in one set of brackets
Line 239: what is this number referring to? Please make it clearer
- Insulin resistance
Line 249: make consistent. Use C26
Lines 253, 254: remove spaces here
Line 268: recommend that (109, 110) citations are placed in one set of brackets
Line 272: recommended change 'due to the...'
Line 272: recommended change ‘favors the tumor's'
Line 276: recommend that (94, 113) citations are placed in one set of brackets
- Bone density
Line 281: should read ‘bone mineral’
Line 284: remove space here
Line 288: remove spaces after (117). And before ‘acting in…’
Line 292: move full stop and list references in one set of brackets (118, 119)
Line 294: what does this suggest? muscle-bone cross-talk? Please elaborate
Line 296: recommend that (122, 123, 124) citations are placed in one set of brackets
Line 302: BMD is bone mineral density, should read whole body bone mineral density
Line 305: should read: post tumor cell inoculation; should read ‘Besides’
Line 307: remove gap before (129)
Line 314: use i.e.
Line 315: remove full stop before . (130).
Line 316: remove full stop before . (131).
- Conclusion
Line 324: recommended modification to improve flow
‘is an illness with tremendous impact, affecting both quality of life and availability of therapeutic options’
Line 325: include ‘multifactorial syndrome’
Line 328: recommend using ‘appropriate’ rather than proper
Line 329: recommend using ‘typically observed weight loss and .....’
Line 338: recommended to incorporate some thinking and clarification around the molecular and cellular cross-talk if you are investigating the immune system as well
Figure 1:
This diagram is interesting and stimulates a lot of thinking. It suggests that all other organ dysfunctions contribute to muscle loss as the tip of the iceberg (the ‘penultimate’ phenotype in cachexia) rather than the extreme processes that are ongoing, in parallel within other organs and tissues.
What about the heart then at the tip of the iceberg as another muscle-type or would this be considered a different metabolic, catabolic tissue? If so, please elaborate why? The heart is included in the diagram but not discussed in the body of the text.
Are the other organs contributors, bystanders or parallel targets of metabolic and molecular changes? What is the author interpretation of this idea?
Which organ or tissue is the strongest trigger in this cascade of pathometabolic / pathocatabolic effects?
Figure 1 Legend:
Line 341: multi-organ dysfunctions are ongoing in parallel during ‘tumor’ … suggest to keep spelling keep consistent
Line 343: include weblink here for drawing software.
Author Response
We thank the reviewer for the positive comments. As suggested, we included in the text more sentences providing a general message/critical comment on the implications of each tissue malfunction (i.e. line 133-136; 198;211; 267-272;). Moreover, we improved the final discussion by providing an author perspective and discussion, thus providing a general message (line 410-425).
Moreover, we are grateful for the list of comments citing the line number. It greatly simplified the revision process. We addressed all the comments and we took the chance for a deeper language editing. For the ones requiring a substantial modification [i.e. Lines as seen in brackets the original line as indicated by the reviewer (38)44;(45)50; (85-100)105-124; (146)200; (182)250; (294)363-367; (338)410-425]. Whenever requested in this list, we elaborated more in details the concept and we distilled the meaning of it. We hope we addressed those concern with sufficient detail.
Figure 1:
This diagram is interesting and stimulates a lot of thinking. It suggests that all other organ dysfunctions contribute to muscle loss as the tip of the iceberg (the ‘penultimate’ phenotype in cachexia) rather than the extreme processes that are ongoing, in parallel within other organs and tissues. What about the heart then at the tip of the iceberg as another muscle-type or would this be considered a different metabolic, catabolic tissue? If so, please elaborate why? The heart is included in the diagram but not discussed in the body of the text. Are the other organs contributors, bystanders or parallel targets of metabolic and molecular changes? What is the author interpretation of this idea? Which organ or tissue is the strongest trigger in this cascade of pathometabolic / pathocatabolic effects?
We are very happy that the figure was stimulating. We actually discussed more in the detail in the conclusion (line 408-412) explaining why we depicted as an iceberg. Actually, concerning heart we realized it was delivering the wrong message adding it at the bottom of the iceberg. Being an emerging issue in cachexia we modified the order of the figures inside the iceberg and we dedicated a few lines to heart in the introduction as an emerging topic. At the bottom of the iceberg, we now inserted liver as we believe (and we wrote in the conclusion) it is the most neglected organ in cachexia (line 416), but likely an important element of wasting as several evidences point out at this organ in the worsening of systemic metabolism.
We kept spelling consistent and we inserted the name of the free repository, we used for figure generation.
Reviewer 3 Report
The review article describes the state of the art about cachexia syndrome, considered at multisystemic level. Overall, the work is well conceived, well written and it provides an useful update in the field.
Author Response
We are grateful to Reviewer 3 for the extremely positive feedback.
Round 2
Reviewer 1 Report
Dear Authors,
Thank you for your revised manuscript. I think that all the comments are responded to and adequately reflected in the revised manuscript. Although there are some limitations in the study design, the authors discussed all biases and weaknesses. I think this manuscript is now acceptable. Thank you for all your efforts.